# Prediction of Critical Buckling Load on Open Cross-Section Columns of Flax/PLA Green Composites

**DOI:** 10.3390/polym14235095

**Published:** 2022-11-23

**Authors:** Liu Jiao-Wang, Sergio Puerta-Hueso, David Pedroche, Carlos Santiuste

**Affiliations:** Department of Continuum Mechanics and Structural Analysis, Universidad Carlos III de Madrid, Avda. de la Universidad 30, 28911 Leganés, Spain

**Keywords:** green composite, open cross-section, buckling

## Abstract

The present work aims to analyze the buckling behavior of nonlinear elastic columns with different open cross-sections and slenderness ratios to verify the limits of the modified Ludwick law to predict the critical buckling load. The results of the analytical formulation based on the modified Ludwick law are compared with a FEM numerical model using the Marlow hyperelastic behavior and experimental results conducted on flax/PLA specimens with three different open cross-sections. The comparative results show that the numerical predictions agree with the experimental results in all the cases. The FEM model can exactly reproduce the buckling behavior of the C-section columns. However, the prediction errors for the C90 and C180 columns are higher than for the C60 columns. Moreover, the theoretical estimations indicate that the C90 cross-section column is the limit of application of the modified Ludwick law to predict the critical buckling load of nonlinear elastic columns with open cross-sections, and the C180 column is out of the prediction limits. Generally, the numerical and theoretical models underestimated the scattering effects of the predictions because more experimental variables were not considered by the models.

## 1. Introduction

In recent years, research into new biodegradable materials, such as natural fibers and bioplastics, has experienced a remarkable expansion thanks to the increasing use of natural resources in industries. Composites reinforced with natural fibers have shown promising properties, including recyclability, lightweight, and low cost. Moreover, they are less hazardous to health and the environment during production and handling than synthetic fibers. The ultimate goal of their development is to replace the use of conventional materials in the industry to the extent applicable. In Europe, many engineers devote financial resources and time to this objective, such as the ended projects “Development of Sustainable Composite materials” [1] and “Sustainable structural and multifunctional biocomposites from hybrid natural fibers and bio-based polymers” [2]. The first one aimed to develop sustainable wood-based composites to substitute fossil-based materials applied in several market sectors. The second focused on developing and optimizing bio-based multifunctional green composites for transportation and other high-value market sectors.

Some potential applications of green composites are automotive components [3], crashworthiness elements to absorb energy [4,5] and orthopedic prostheses [6]. Moreover, compared with synthetic fibers such as glass and carbon fibers, natural fiber presents good behavior as a thermal and acoustic insulator [7,8], and it has the potential to be applied in critical applications that involve tensile and cyclic loading [9]. Despite the promising applications of green composites in the industry, they face several severe drawbacks due to their material properties, such as poor fire resistance, high moisture absorption, and lower durability. Modification in the composition and/or the manufacture of these materials becomes an effective way to remedy these weaknesses of the potential material, such as partially mixing with synthetic fibers or with the surface treatments of the reinforcing fibers. However, this action undermines the overall environmentalism of the material.

An additional problem with green composites is their complex mechanical behavior. While traditional composites reinforced with synthetic fibers used to be modeled as linear-elastic up to failure, the mechanical behavior of green composites presents nonlinear elasticity, viscous effects, and plastic behavior before failure [10,11]. This feature requires unconventional constitutive models to define this material. Some frequently used theoretical models are the Ramberg–Osgood theory and the Ludwick-type theory. The modified Ludwick law [12] was developed to predict the mechanical behavior of the flax fiber-reinforced polymer composite in nonlinear buckling problems. Moreover, it was experimentally validated for the first time to effectively predict the buckling behavior of the nonlinear elastic column [13].

Another technique to predict the nonlinear elastic behavior of natural fiber-reinforced composites (NFRP) is the Finite Element Method (FEM). Some typically used software are ANSYS and ABAQUS. Specifically, various nonlinear elastic models are available for use in the library of Abaqus to model the nonlinear elastic stress–strain response of the NFRP that resembles the hyperelastic behavior.

The constitutive hyperelastic models assume instantaneous elastic responses up to large strains and are often incompressible. The material state does not depend on the history or the strain rate. The hyperelastic material model is, thus, defined as a strain energy potential. There are several forms of strain energy potential available in Abaqus and they are generally categorized into two types: physically motivated, which studies the material response from a microstructural point of view and phenomenologically motivated, which starts from a continuum point of view. Some models are typically used to study nonlinear elastic materials, such as Arruda–Boyce, Van der Waals, Yeoh, and Marlow [14].

Some studies published in the open literature that implicate these material models are [15,16,17]. Shahzad et al. [15] evaluated five hyperelastic material models in Abaqus to predict the mechanical behavior of carbon-black-reinforced natural rubber under uniaxial, biaxial, planar, and volumetric tests. The experimental test data were used to calibrate these model’s coefficients and the comparative analysis showed that the Yeoh model presented the best fit over other material models. Tobajas et al. [16] carefully compared five constitutive models to reproduce the mechanical behavior of an elastomer thermoplastic material, Santoprene 101-73. Considering only the uniaxial test data, they demonstrated that the Marlow model showed the best accuracy compared to the others.

Anandan et al. [17] studied the mechanical response of an oil palm shell-reinforced rubber composite under repetitive compression loads using experimental tests and numerical models. Specimens with different shapes were fabricated and experimentally tested and then hyperelastic models in Abaqus software were created to reproduce the stress–strain behavior of the material. The results found that the Yeoh model better suits all the geometries, followed by the Marlow model.

Phenomenologically motivated models such as Yeoh and Marlow work well and are less complex than physically motivated ones. They are frequently used to simulate the mechanical behavior of hyperelastic materials for a wide range of strains and deformation modes, especially when extensive test data are unavailable [18,19]. Nonetheless, the Yeoh form involves three deviatoric and three volumetric coefficients in its formulation, while the Marlow form does not require a parameter to be fit [14,20,21]. Therefore, when only limited test data are available, using a more extensive form, the Yeoh model can increase the instability of the numerical model due to the higher number of involved parameters. Thus, the Marlow model becomes the most straightforward hyperelastic material model to define the stress–strain behavior of NFRP composites [22].

The nonlinear elastic behavior of green composites implies that the prediction of buckling behavior is more complex than that in traditional composites reinforced with synthetic fibers. There are several works focused on the study of the buckling behavior of nonlinear elastic columns but the number of works using green composites is quite reduced. Most of them are theoretical works that assume a nonlinear elastic behavior following the Ludwick law as in [23]. Lee and Lee [23] analyzed the buckling response of tapered rectangular columns that obey the Ludwick-type material behavior. They developed a mathematical model and solved it with two FORTRAN computer programs. However, experiments were not conducted. According to them, the numerical model, assuming a linear elastic square-profile steel column, was validated with the analytical results obtained in two previous works [24,25]. The comparative study of the rectangular profile cantilever column showed excellent agreement but they also expressed that the analysis should be carried out in columns with different cross-section forms.

Gopalan et al. [26] studied the critical buckling load of a woven flax/bio-epoxy composite (WFBE) under a uniaxial compression test using experimental, numerical, and analytical techniques. First, the numerical model was validated using the experimental results, and then, a regression equation was created based on the numerical results. The comparison showed that the numerical predictions obtained using ANSYS16 were in good agreement with the experimental results. The analytical results from the regression equation showed an error between 3 and 5% compared to the numerical results. Subsequently, numerical, and analytical approaches were used to estimate the critical buckling load of different WFBE specimens by varying the number of layers, the width of the plate, and the ply orientation. As a result, they concluded that the number of layers was more influential than the remaining factors.

Slender composite columns with open cross-sections are less frequently studied because it is a more complex buckling problem. Local buckling is more likely to happen for open-section structures than closed-section ones due to reduced structural stability and torsional rigidity. This phenomenon depends on the element’s geometry, boundary, and loading condition [27]. Although the study performed by these authors is not concerned with a natural fiber-reinforced polymer composite, the same problem can occur for green composite columns as the failure is not mainly caused by the material strength. Rahima Shabeen and Rajesh [28] conducted an experimental and numerical study on the influence of the geometric parameters on the buckling behavior of an L-section column, which was made of E-glass fiber-reinforced polyester isophthalic resin. They found long columns failed because of general buckling but the shortest specimen presented torsional buckling near the loading end under uniaxial compression. Moreover, they observed that the buckling strength decreases with the increase in the slenderness ratio and the width-to-thickness ratio.

The comparison between experimental tests and theoretical predictions is difficult because experimental results are scattered due to different reasons: uncertainty in data collection, problems in the manufacture and cutting of the specimens and sensitivity to environmental factors such as temperature and humidity. These problems are intensified in the case of natural materials because their properties are much more variable and depend on other factors such as state of conservation, harvest conditions and growing region. The modified Ludwick constitutive model was experimentally validated in predicting the critical buckling load of nonlinear elastic columns in previous work [13]. As a result, the analytical and numerical estimations of the critical buckling load of rectangular green composite columns predicted accurately the experimental results. However, the use of modified Ludwick law was only applied to rectangular columns with high slenderness.

The present work is a continuation of the previous study but analyzes different cross-sections to verify the limits of the Ludwick law in the predictions of critical buckling load. Columns with an open C-section were studied and three arc angles were chosen to manufacture columns with different slenderness ratios using flax/PLA (polylactic acid) green composites. The validity of an analytical formulation developed from the modified Ludwick law and a numerical model using the Marlow hyperelastic model are verified by comparison with the experimental results.

## 2. Materials and Methods

### 2.1. Methodology

The main goal of this study is to find the limits of the validity of the modified Ludwick law to predict the critical buckling load of nonlinear columns. The study hypothesizes that modified Ludwick law can be used in an analytical model to reproduce the nonlinear elastic behavior of green composites and Euler theory to define the critical buckling load is only valid for slender columns. The buckling modes cannot be predicted using a beam theory in columns with a low slenderness ratio. Therefore, three different geometries are chosen to manufacture beams with slenderness ratios between 48.89 and 382.66. The analysis combines theoretical formulation, numerical modeling, and experimental tests:(1)Characterization tests. Flat rectangular specimens were manufactured and subjected to a uniaxial tensile test to characterize the nonlinear behavior of flax/PLA green composites. Results of the characterization tests showed significant scattering in the stress–strain curves; thus, three different curves were selected to consider the intrinsic variation of natural fiber-reinforced composites;(2)Theoretical formulation. The theoretical model is based on the work of Saetiew and Chucheepsakul [29,30], which used the modified Ludwick law to reproduce the nonlinear elastic behavior of material to predict the critical buckling load of columns. In this work, this formulation is applied to open C-sections;(3)Numerical modeling. A numerical model was developed in Abaqus/Standard commercial software using Marlow hyperelastic model to predict the critical buckling loads;(4)Experimental tests. Open C-sections with three different geometries were manufactured using the compression molding method with flax/PLA green composites. Six specimens of each geometry were manufactured and tested to evaluate the critical buckling load;(5)Analysis of the results. For each cross-section, the theoretical and numerical models provided three results because the three different stress–strain curves were used in each model. Thus, the scattering of the experimental results was reproduced by the models. The values of critical buckling loads and buckling modes of theoretical and numerical modes were compared with experimental results.

### 2.2. Theoretical Approach

Between the nonlinear elastic theories, modified Ludwick constitutive law is proved effective in characterizing the tensile response of flax fiber-reinforced biopolymer composite [13]. This law is also an enhanced version of the original Ludwick law by remedying the defect of the stress slope going to zero or infinity when the deformation (ε) approaches 0 of the original Ludwick formulation, see Figure 1.

The additional term in Equation (1), *ε*_0_ is a new elastic constant that is used to avoid this problem and makes further calculations feasible. The remaining parameters of the equation are *E* is the Young modulus, and *n*, is the material’s nonlinearity. These three parameters are obtained using the least square fitting method to fit the experimental results of the stress–strain curves of the composite as initial data.
(1)$$\sigma =\{\begin{array}{c} E\cdot \left\{{\left(\varepsilon +\varepsilon_{0}\right)}^{\frac{1}{n}}-\varepsilon_{0}{}^{\frac{1}{n}}\right\};\;\;\;\;\;\;\varepsilon \geq 0,\\ -E\cdot \left\{{\left(-\varepsilon +\varepsilon_{0}\right)}^{\frac{1}{n}}-\varepsilon_{0}{}^{\frac{1}{n}}\right\};\;\;\;\;\;\;\varepsilon <0. \end{array}$$

The critical buckling load of a column with pinned–pinned boundary conditions comes from taking the moment about the loaded point,
(2)M=−P·ϑ
and considering the differential equation that describes the deflection curve,
(3)$$E\cdot I_{o}\cdot \frac{d^{2}\vartheta }{dx^{2}}=M$$
where *x* is the longitudinal coordinate; *M* is the value of the bending moment; $E\cdot I_{o}$ are the Young modulus and second moment of inertia; *P* is the axial load; and ϑ is the transverse displacement of the column, as can be seen in Figure 2. Subsequently, by substituting Equation (2) into Equation (3) and solving the linear, homogenous, second-order differential equation yields, the following general expression with two constants *C*_1_ and *C*_2_, is achieved for Equation (4).
(4)$$C_{1}\cdot \sin\left(\sqrt{\frac{P}{E\cdot I_{o}}}\cdot x\right)+C_{2}\cdot \cos\left(\sqrt{\frac{P}{E\cdot I_{o}}}\cdot x\right)=\vartheta \left(x\right)$$

Two boundary conditions are presented:
The first one is that the transverse deflection ϑ(0) is zero at *x* = 0 and then the solution of Equation (4) is *C*_2_ = 0;Second, the transverse deflection ϑ(L) is zero at *x* = *L* and there are two solutions for the constant *C*_1_:

Solution 1: The trivial solution, i.e., *C*_1_ = 0;

Solution 2: the set of values when $\sin\left(\sqrt{\frac{P}{E\cdot I_{o}}}\cdot L\right)=0,$ i.e., $\left(\sqrt{\frac{P}{E\cdot I_{o}}}\cdot L\right)=0,\;\pi ,\;2\pi ,\ldots ,n\pi$.

Therefore, being *n* = 1, the final expression for the critical buckling load has the form of Equation (5).
(5)$$P_{cr}=\frac{\pi^{2}\cdot E^{\prime }\cdot I_{o}}{L^{2}}$$

Using Equation (1), the modified Ludwick Law, and according to the statement of Saetiew and Chucheepsakul [29] that for small values of strains, most elastic material with nonlinear behavior can be approximated to linear elastic materials, the analytical approximation of the critical buckling load, Equation (6), was developed for a pinned–pinned nonlinear elastic column, assuming small deformations.
(6)$$P_{cr}=\frac{\pi^{2}\cdot E\cdot I_{o}}{L^{2}}\cdot \left(\frac{\varepsilon_{0}{}^{\frac{1-n}{n}}}{n}\right)$$
where *L* is the column length and *E*, *n* and *ε*_0_ are the parameters of the modified Ludwick law [13].

Three curved cross-sections, whose arc angles vary from 60 degrees to 180 degrees, were analyzed in this study, see Figure 3. Their moment of inertia expressions along the *x*-axis, using Equations (7) to (9), were developed by integrating the area moment of inertia for the ring and then applying the Parallel Axis Theorem to get the moment of inertia at the centroidal axis of the section.
C-section 180°
(7)$$I_{xc}=R^{3}\cdot t\cdot \left(\frac{\pi }{2}-\frac{4}{\pi }\right)$$
2.C-section 90°
(8)$$I_{xc}=R^{3}\cdot t\cdot \left(\frac{\pi }{4}+\frac{1}{2}-\frac{4}{\pi }\right)$$
3.C-section 60°
(9)$$I_{xc}=R^{3}\cdot t\cdot \left(\frac{\pi }{6}+\frac{\sqrt{3}}{4}-\frac{3}{\pi }\right)$$
where *R* is the medium radius and *t* is the thickness of the ring section.

### 2.3. Experimental Setup

The specimens are made of two components: woven flax fibers without chemical treatment (areal density and thickness are 463.3 g/m^2^ and 0.94 mm, respectively), and polylactic acid pellets from NaturePlast (France), commercial name of PLI005 (density of 1.24 g/cm^3^ according to ISO 1183, melting temperature of 175 °C, Young Modulus of 3500 MPa, and tensile strength at break of 50 MPa).

The fully biodegradable composite plates were manufactured by compression molding. Pressure was applied by a universal testing machine and temperature by two thermo-heating plates [13]. First, the flax fabrics and the PLA pellets were previously dried in the oven at 65 °C for half an hour to remove humidity. Then, 12.3 g of PLA pellets were heated and pressed in the universal machine under 180 °C for about two minutes to melt into a rectangular film of 200 × 160 mm^2^. Figure 4a shows the dimensions of the universal testing machine with two heated plates. Next, four layers of flax fabrics and five layers of PLA films were alternatively stacked to make a plate of 200 × 260 mm^2^ under 180 °C and 16MPa for five minutes (including two minutes of preheating). Finally, the compressed biocomposite plate was removed from the machine and quickly placed into the metal molds, see Figure 4b,c. The plate acquired the mold form after completely cooling to room temperature. Then, it was cut into three types of specimens: C-section 180 °C-section 90°, and C-section 60°, with the simplified nomenclature: C180, C90, and C60, respectively, Figure 4d.

Uniaxial tensile tests were carried out on (120 × 20 mm^2^) rectangular flat specimens of the biodegradable composite in a universal testing machine under quasi-static conditions (speed of 0.3 mm/min) to characterize the non-linear elastic behavior of the composite. Then, a pronounced scattering effect was observed in the stress–strain diagrams, Figure 5. At least 24 specimens of the same dimension and material were tested in the laboratory, and the material strength tendency of each one varied within the specific range. The six most representative curves are shown in the figure below, from the weakest case to the strongest one (C4S1 to C4S6).

To ensure the accuracy of further studies, three stress–strain curves were selected to characterize the mechanical response of the biodegradable composite, and they were: the highest (C4S2), the lowest (C4S6), and an intermediate curve (C4S1). The three parameters *E*, *n*, and $\varepsilon_{0}$ of the modified Ludwick constitutive law were fitted using least square method for the three materials and they are listed in Table 1.

Figure 6 shows the comparison between numerical data and modified Ludwick law results for the stress–strain curves. An excellent agreement can be seen in the three tests. Therefore, these values were introduced into the equation to calculate the theoretical critical buckling load of the columns and they are used as the input uniaxial test data for the Marlow hyperelastic model in Abaqus.

The buckling test setup is shown in Figure 7. Two parallel plates compressed the specimen in a universal testing machine (INSTRON 8516) at a constant velocity of 0.3 mm/min to reproduce quasi-static conditions. During the buckling test, the cross-sections in contact with the compression plates did not experience relative displacement with the plate due to friction forces but no clamping device was used, therefore the cross-section rotation was not restricted.

### 2.4. FEM Model

As natural fiber-reinforced biopolymer composites present nonlinear elastic behavior, their stress–strain diagram can be assimilated as elastomeric materials and be modeled using the hyperelasticity material model. In Abaqus simulation software, several hyperelastic strain energy potentials are available for use. Some of them are the reduced polynomial model, the Arruda–Boyce model, and the Marlow model [14].

The effectiveness of these models in matching the test data of green composites was compared and analyzed in previous work [13]. As a result, the Marlow model was selected to reproduce the nonlinear elastic behavior of the green composite, and its strain energy potential formulation is the following:(10)$$U=U_{dev}\left(\bar{I_{1}}\right)+U_{vol}\left(J_{el}\right)$$
where $U_{dev}$ is the deviatoric potential energy and $U_{vol}$ is the volumetric potential energy that depends on the elastic volume ratio $J_{el}$. To reduce the complexity of the modeling, the volumetric behavior of the material is not defined, and then Abaqus considers an incompressible material model. Subsequently, when the Marlow model is only dependent on the first term $U_{dev}\left(\bar{I_{1}}\right)$, being $\bar{I_{1}}$ the first deviatoric strain invariant, the following expression of the strain–energy density function is obtained [20]:(11)$$U\left(I\right)=\int_{0}^{\lambda_{T}\left(I\right)-1}T\left(\varepsilon \right)\;d\varepsilon$$
where T(ε) is the value of uniaxial stress as a function of uniaxial strain obtained in a uniaxial tensile test, while $\lambda_{T}$ is the value of the stretching.

Moreover, when only limited test data are available, using more extensive models (such as polynomial models) can increase the instability of the numerical model due to the higher number of involved parameters in the complex models. Thus, the Marlow model becomes the most appropriate and most straightforward hyperelastic material model to define the stress–strain behavior of the flax/PLA composite.

Biocomposite specimens were modeled in Abaqus/Standard as the solid deformable body. They obey the isotropic Marlow hyperelastic material model, and the experimental data obtained in the uniaxial tensile test were provided as the input. The boundary conditions were pinned–pinned to reproduce the experimental setup where the cross-section rotation was not restricted.

The results of a mesh sensitivity analysis conducted to check the influence of mesh size are shown in Figure 8, all the meshes were uniform. The optimal mesh for an adequate balance between the accuracy of the results and the computational cost has approximately 4400 linear hexahedral elements with reduced integration (C3D8R in Abaqus) and 9246 nodes. Moreover, there was a total running time of 40 s.

## 3. Results and Discussion

The results analyzed in this section include experimental tests and predictions from the numerical model and analytical approach. A total of 18 experimental buckling tests were carried out because six specimens were manufactured for each cross-section. The six specimens of each configuration had slight differences in thickness and length; thus, the numerical model and analytical approach were adapted for each specimen. Moreover, three different stress–strain curves were included in the numerical and analytical models for each specimen to reproduce the variability of the composite properties. All the results are listed in Table 2, including the critical load obtained in experimental tests, theoretical approach, and numerical model for the 18 specimens and their dimensions.

Figure 9 compares the buckling modes predicted by the FEM model and the experimental results. C60 and C90 columns, Figure 9a,b, respectively, showed the classical buckling mode of slender columns dominated by the bending moment of the cross-section. However, the C180 column showed a lateral buckling mode that the analytical approach could not predict. Although only a compression load in the axial direction was applied, the first buckling mode of the C180 column was lateral buckling with a clear rotation of the cross-section. Both phenomena, buckling due to bending and lateral buckling, are instability conditions that appear when a compression load is applied, but classical bending buckling is the first buckling mode and it is only for slender beams. In the three cases, there is an excellent agreement between the numerical predictions and experimental results.

Figure 10 shows the results of the critical buckling load obtained in the experimental tests and compared with the numerical and theoretical predictions for the C60 cross-section. The results are represented in box and whisker charts showing the data distribution graphically, allowing for a more straightforward statistical analysis of the problem. The top and bottom lines represent the maximum and minimum values, the box limits indicate the upper quartile (Q3) and lower quartile (Q1), the midline represents the median, and the *X* mark the mean value.

The comparison between the experimental, numerical, and theoretical results shows an excellent agreement. It indicates that both numerical and theoretical approaches can reproduce the buckling behavior of nonlinear elastic columns with a slenderness ratio of about 378. The three average values are quite similar (8.15 N, 8.18 N, and 8.02 N for experimental, theoretical, and numerical results, respectively), the error of the theoretical approach is 0.41%, and the numerical model is 1.59%. Moreover, the scattering of the experimental results is also reproduced by the numerical and theoretical models. The minimum and maximum values of the experimental tests are 6 N and 9.9 N. The FEM predictions are 6.33 N and 9.76 N for minimum and maximum loads; thus, the errors were 5.43% and 1.37%. At the same time, the theoretical model predicted a minimum critical buckling load of 6.91 N, an error of 15.15%, and a maximum of 9.45 N, an error of 4.52%.

On the other hand, the variability of the results was included in the models taking into account the variation of mechanical properties observed in tensile tests and the geometry of the specimens. However, both numerical and theoretical models underestimated the scattering because there were more experimental parameters that affected the scattering of the results that were not considered by the models.

The comparison of the experimental, numerical, and theoretical results in critical buckling load for the C90 cross-section is represented graphically in Figure 11. The slenderness ratio of the C90 columns ranges between 170 and 173, which is relatively lower than that of the C60 columns. The agreement between the experimental results and numerical and theoretical model predictions is reasonable but clearly worse than for the C60 cross-section.

The average experimental result of C90 is 49.70 N, while the theoretical mean value is 60.32 N and the numerical mean load is 65.42 N; the theoretical error is 21.36% and the numerical error is 31.63%. However, the results scattering was predicted with reasonable accuracy since the difference between the maximum and minimum values was 18.50 N, 16.67 N, and 22.67 N for the experimental, theoretical, and numerical results, respectively.

Both models overestimated the critical buckling load in the C90 cross-section columns. Furthermore, the FEM predictions always show higher values than the theoretical method. Figure 9b compares the three buckling modes, showing that the numerical model reproduces the same deformed shape as experimental tests. Therefore, the C90 cross-section column with a slenderness ratio of about 170 is the limit of application of the modified Ludwick law to predict the critical buckling load of nonlinear elastic columns with open cross-sections. The buckling mode is properly predicted but the error in the value of the critical buckling load is considerable.

Figure 12 shows the results of the critical buckling load obtained in the experimental tests, numerical model, and theoretical approach for C180 cross-section columns. It should be observed that in this case, the boundary conditions of the FEM model were modified to clamped–clamped to reproduce the buckling failure mode observed in experimental tests. When the buckling mode was dominated by the cross-section bending, as in the C60 and C90 columns, the pinned–pinned boundary conditions reproduced the experimental setup because there was no clamping device to avoid the cross-section bending rotation that separated some points of the end cross-sections. However, the buckling mode of the C180 column was dominated by lateral buckling with a torsional rotation of the cross-section. The friction forces between compression plates and the specimens could not avoid the bending rotation. However, they prevented the torsional rotation of the cross-section because they acted in the contact plane. Thus, the boundary condition must be clamped–clamped to reproduce the lateral buckling mode.

The comparative analysis shows that the analytical results are distant from the experimental data; not even the variability outside the lower quartiles is close to the experimental data. The average experimental critical buckling load is 599.17 N, while the average theoretical prediction is 1462.06 N. The boxplot shows that the theoretical predictions have a much larger average critical buckling load than the experimental results. However, the numerical approach does show better results. The average prediction of the FEM model for the C180 critical buckling load is 727.93, with an error of 21.49%. The mean error is very high but considering the dispersion of the results, there is a superposition between numerical and experimental results. The average experimental result (599.17 N) is close to the minimum numerical prediction (579.49 N) and the maximum experimental data (730 N) is near the average numerical prediction (727.93 N).

The buckling mode observed in experimental tests and numerical simulations, Figure 9c, is clearly dominated by lateral torsional buckling. This buckling mode can be reproduced by the FEM model but it is not included in the theoretical formulation. Thus, the C180 cross-section columns, with a slenderness ratio between 48 and 52, are out of the limits of the modified Ludwick law method to predict the critical buckling load of nonlinear elastic columns. The prediction of the critical buckling load in nonlinear elastic columns with low slenderness ratios requires the inclusion of lateral torsional buckling in the formulation because this buckling load is lower than the bending buckling load included in the present formulation. A comparison of the three cross-sections is summarized in Table 3. The maximum, minimum and average results of the experimental tests, theoretical approach and numerical model are summarized. This comparison shows that the C180 cross-section columns, with slenderness ratios of about 50, are not suitable for the use of the modified Ludwick law when predicting the critical buckling load. On the other hand, the modified Ludwick law was used successfully to predict the critical buckling load of the C90 cross-section columns, with slenderness ratios of about 170. The relatively high errors in the C90 columns indicate that the limit of application must be near 170 slenderness ratios but more experiments are necessary to find this limit with more accuracy.

## 4. Conclusions

In this study, the limits of the application of the modified Ludwick law to predict the critical buckling load of columns with nonlinear elastic behavior are investigated. The analytical model based on the modified Ludwick law is compared with a FEM model including the Marlow hyperelastic model in the formulation and experimental tests on specimens manufactured with PLA reinforced with flax woven fibers. To find the Ludwick law limits, C-section columns with different slenderness ratios are analyzed. The main conclusions of the comparison are:There is an excellent agreement between numerical predictions and experimental results in the three cases. The FEM model can exactly predict the buckling behavior of the C-section columns. Even the lateral buckling mode of the C180 columns was reproduced by the numerical model;The numerical error that used the Marlow hyperelastic model for the C60 columns is only 1.59%, for C90 it is 31.63%, and for C180 it is 21.49%. The buckling mode of these three cases was predicted correctly but the errors in the value of critical buckling load for C90 and C180 are considerable;For columns as slender as C60, the critical buckling load formulation based on the modified Ludwick law matches perfectly with the experimental results with an error of only 0.41%. For the C90 columns, this error reaches 21.36%, and for C180, the error is unacceptably high. Therefore, the C90 cross-section column with a slenderness ratio of about 170 is the limit of application of the modified Ludwick law to predict the critical buckling load of nonlinear elastic columns with open cross-sections, and the C180 column is out of prediction limits;Moreover, the theoretical prediction of the critical buckling load in nonlinear elastic columns with low slenderness ratios requires the inclusion of lateral torsional buckling in the formulation because this buckling load is lower than the bending buckling load included in the present formulation;Generally, the numerical and theoretical models underestimated the scattering effects of the predictions because more experimental variables were not considered by the models, especially in the analytical model.

## Figures and Tables

**Figure 1 polymers-14-05095-f001:**
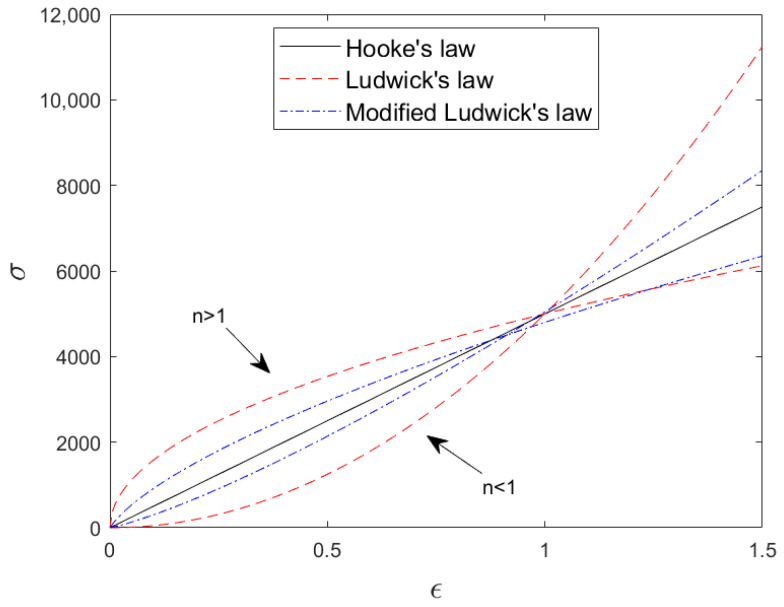
Stress–strain diagram of typical Hooke’s law, Ludwick Constitutive law, and modified Ludwick law.

**Figure 2 polymers-14-05095-f002:**
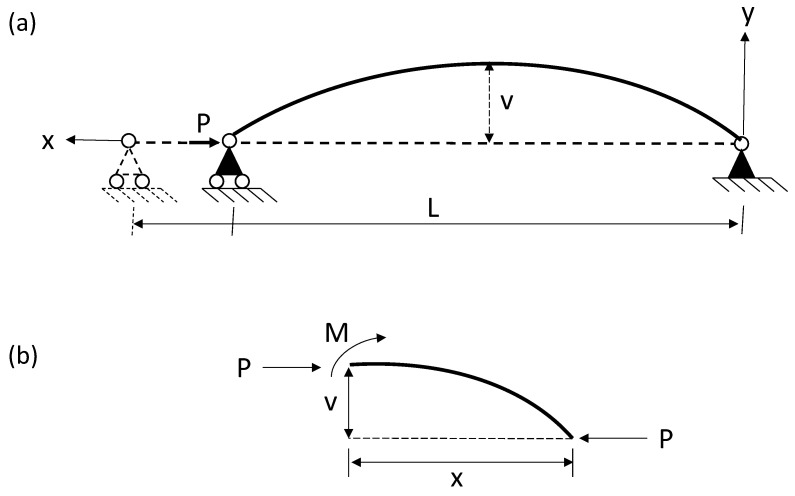
Column under uniaxial compression load with pinned–pinned boundary conditions, (**a**) deformed shape, (**b**) free-body diagram. *M*: the bending moment; *x* the longitudinal coordinate; *P* the axial load; and *v* the transverse displacement of the column.

**Figure 3 polymers-14-05095-f003:**
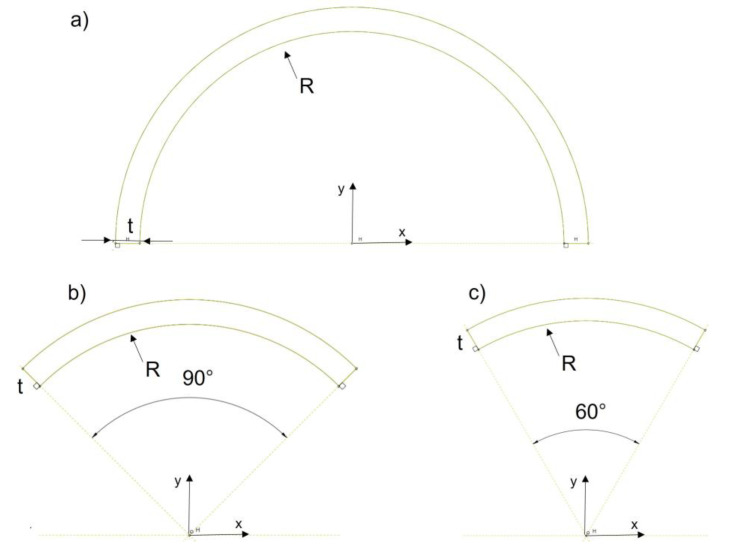
Cross-sections of the columns, (**a**) 180 °C-section, (**b**) 90 °C-section, (**c**) 60 °C-section.

**Figure 4 polymers-14-05095-f004:**
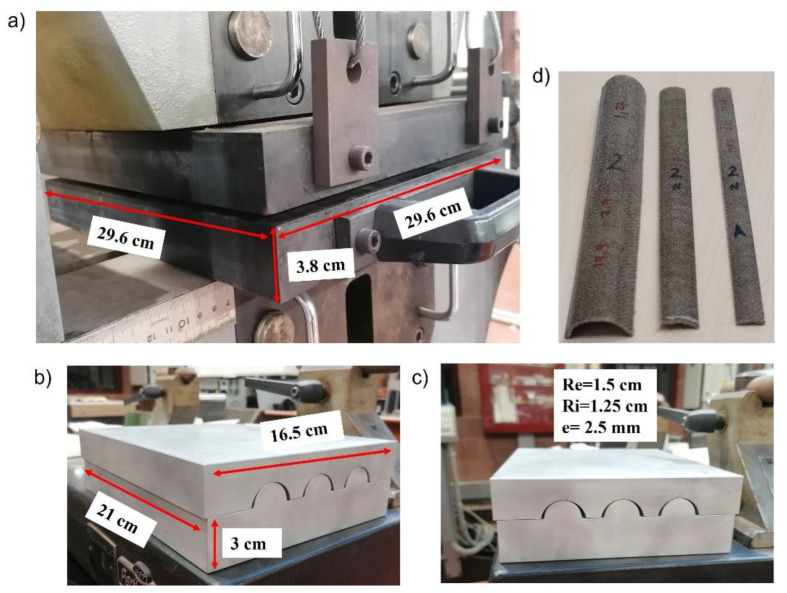
(**a**) Universal testing machine and heating plates, (**b**,**c**) Metallic molds for C-section columns, (**d**) manufactured specimens with three different cross-sections.

**Figure 5 polymers-14-05095-f005:**
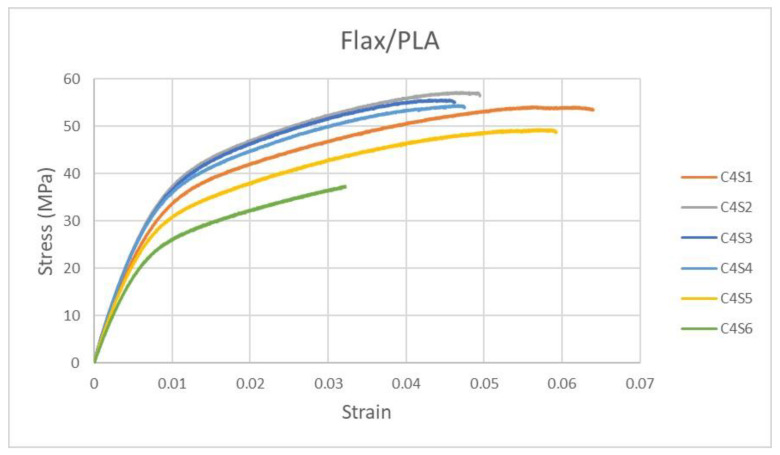
Stress–strain curves of flax/PLA composite.

**Figure 6 polymers-14-05095-f006:**
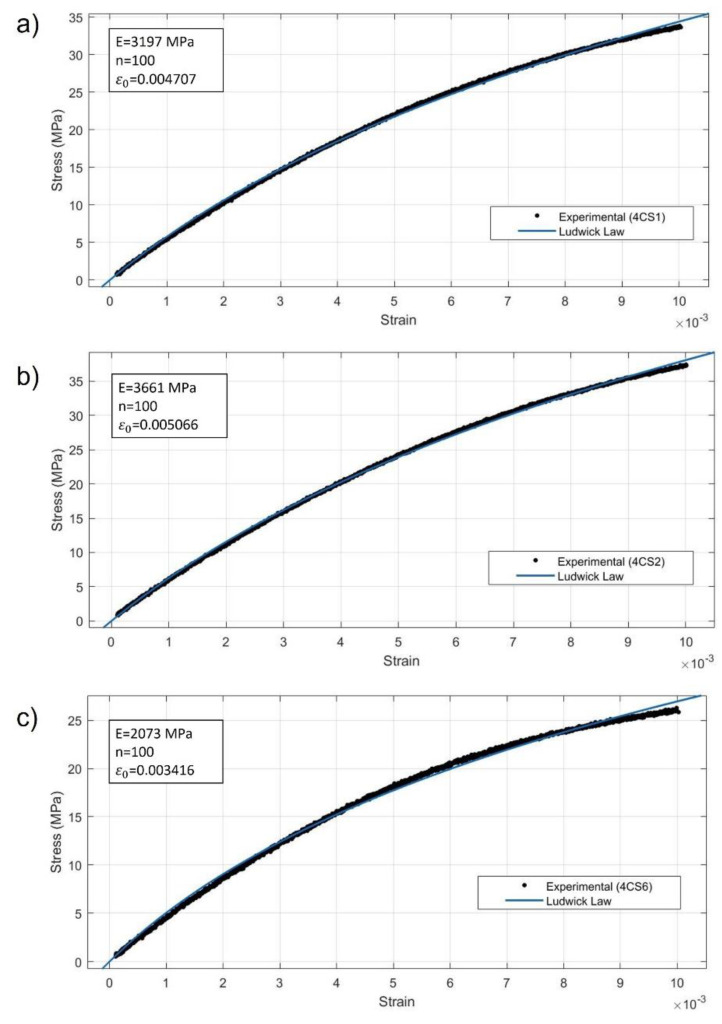
Stress–strain curves of the three representative specimens: (**a**) specimen number 4CS1, (**b**) specimen 4CS2, and (**c**) specimen 4CS6. Curve fitting using modified Ludwick law.

**Figure 7 polymers-14-05095-f007:**
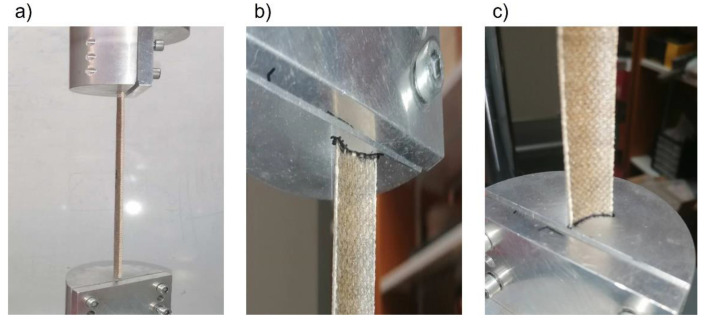
Buckling test of C90 column: (**a**) general view, (**b**) detail of upper support, (**c**) detail of bottom support.

**Figure 8 polymers-14-05095-f008:**
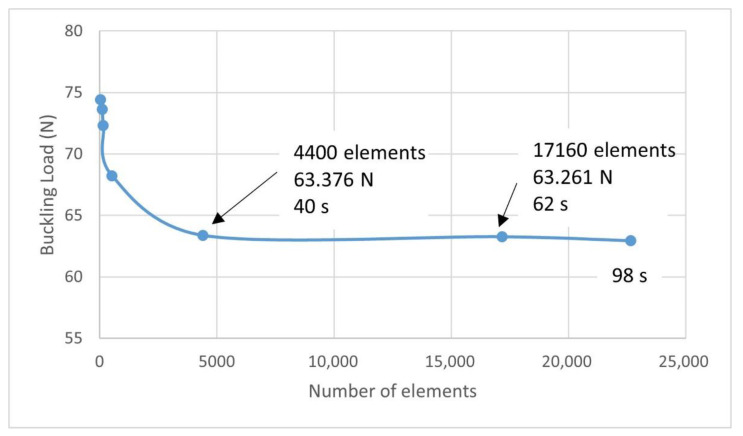
Simulation in Abaqus of the buckling test. Mesh sensibility analysis.

**Figure 9 polymers-14-05095-f009:**
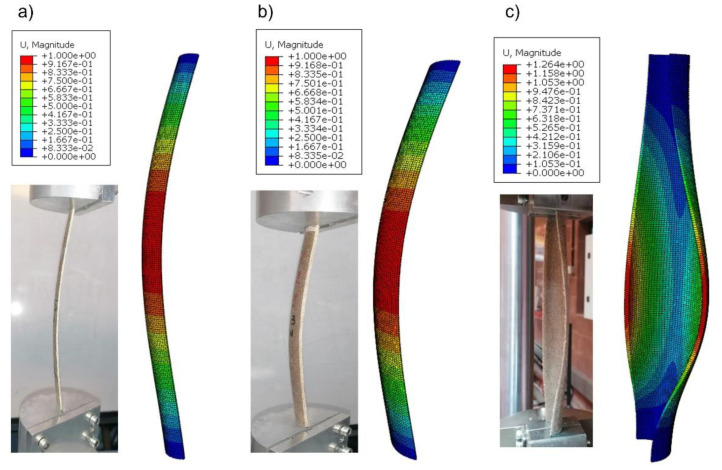
Buckling modes, comparison between experimental results and numerical predictions: (**a**) C60 section, (**b**) C90 section, (**c**) C180 section.

**Figure 10 polymers-14-05095-f010:**
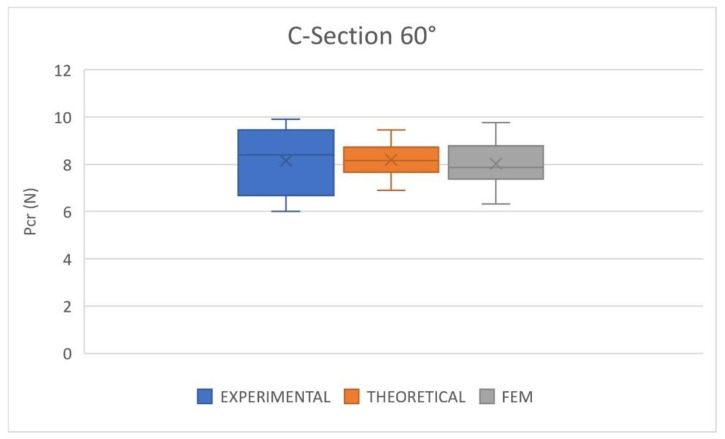
Critical buckling load. Comparison between experimental, theoretical and FEM results. C60 cross-section.

**Figure 11 polymers-14-05095-f011:**
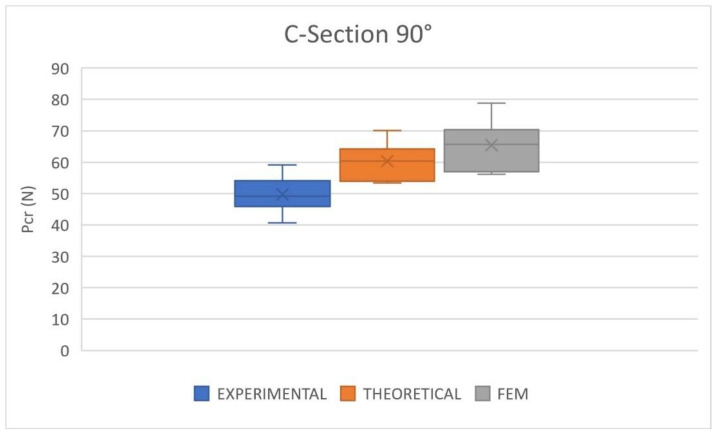
Critical buckling load. Comparison between experimental, theoretical and FEM results. C90 cross-section.

**Figure 12 polymers-14-05095-f012:**
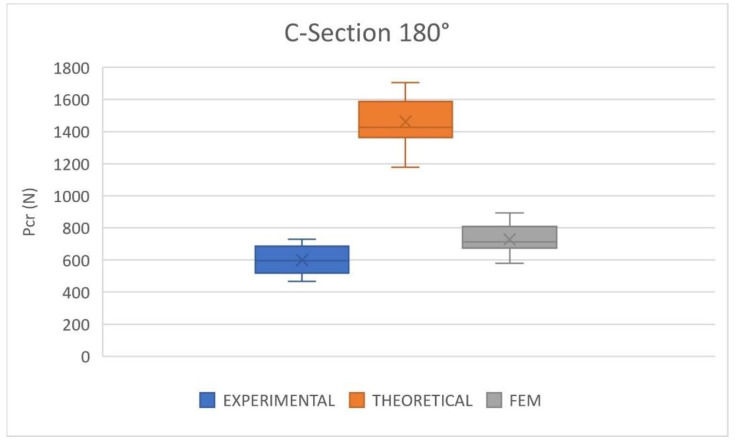
Critical buckling load. Comparison between experimental, theoretical and FEM results. C180 cross-section.

**Table 1 polymers-14-05095-t001:** Mechanical properties of the nonlinear elastic columns.

	*E*	*n*	$\varepsilon_{0}$	*E*-Square
C4S1	3197	100	0.004707	0.9991
C4S2	3661	100	0.005066	0.9992
C4S6	2073	100	0.003416	0.997

**Table 2 polymers-14-05095-t002:** Critical buckling loads obtained from the three approaches: experimental tests, numerical model and analytical approach.

Specimen	Length (mm)	Thickness (mm)	M. Inertia (mm4)	Slenderness	Pcr exp	Pcr the_ C4S1	Pcr the_ C4S2	Pcr the_ C4S6	Pcr FEM_ C4S1	Pcr FEM_ C4S2	Pcr FEM_ C4S6
C180_1	200.00	1.33	899.09	49.37	542.00	1428.13	1520.62	1271.91	687.27	736.47	592.26
C180_2	199.50	1.32	891.28	49.27	535.00	1422.84	1514.99	1267.20	679.63	728.28	585.68
C180_3	199.00	1.45	997.99	48.89	672.00	1601.20	1704.90	1426.05	820.78	879.53	707.31
C180_4	210.00	1.35	918.66	51.79	467.00	1323.56	1409.28	1178.78	672.45	720.58	579.49
C180_5	200.00	1.42	974.02	49.19	649.00	1547.15	1647.36	1377.92	783.82	893.93	675.46
C180_6	199.00	1.44	985.98	48.92	730.00	1581.94	1684.40	1408.90	804.46	862.04	693.25
C90_1	199.00	1.35	37.38	171.71	52.30	59.97	63.85	53.41	65.14	69.80	56.14
C90_2	199.00	1.35	37.54	171.68	59.20	60.23	64.13	53.64	65.14	69.80	56.14
C90_3	200.00	1.36	37.86	172.48	47.90	60.14	64.03	53.56	65.59	70.28	56.52
C90_4	198.00	1.35	37.38	170.85	50.50	60.58	64.50	53.95	65.75	70.46	56.66
C90_5	200.00	1.37	38.18	172.41	40.70	60.65	64.58	54.01	66.29	71.03	57.12
C90_6	200.00	1.47	41.44	171.76	47.60	65.82	70.08	58.62	73.54	78.81	63.38
C60_1	202.50	1.41	5.46	382.66	6.00	8.46	9.01	7.53	8.66	9.28	7.47
C60_2	198.50	1.42	5.51	374.96	6.90	8.88	9.45	7.91	9.11	9.76	7.85
C60_3	196.00	1.36	5.24	371.08	8.40	8.66	9.22	7.71	8.57	9.18	7.38
C60_4	200.00	1.28	4.88	379.80	9.30	7.76	8.26	6.91	7.34	7.87	6.33
C60_5	200.00	1.32	5.06	379.23	9.90	8.04	8.56	7.16	7.79	8.35	6.71
C60_6	199.00	1.31	5.02	377.47	8.40	8.05	8.57	7.17	7.74	8.30	6.67

**Table 3 polymers-14-05095-t003:** Comparative results of the three cross-sections. Including average, maximum, and minimum results of experimental tests, FEM model, and theoretical approach. Unit in N.

		C60	C90	C180
Experimental	Maximum	9.90	59.20	730.00
	Average	8.15	49.70	599.17
	Minimum	6.00	40.70	467.00
Theoretical	Maximum	9.45	70.08	1704.90
	Average	8.18	60.32	1462.06
	Minimum	6.91	53.41	1178.78
FEM	Maximum	9.76	78.81	893.93
	Average	8.02	65.42	727.93
	Minimum	6.33	56.14	579.49

## Data Availability

Not applicable.

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
