# Peer review of "Prediction of Critical Buckling Load on Open Cross-Section Columns of Flax/PLA Green Composites"

_polymers, 2022, doi:10.3390/polym14235095_

Round 1

Reviewer 1 Report

The submitted manuscript focuses on an interesting and topical subject. In my opinion, the topic of the paper has more potential than the authors have exploited. The article is publishable after major modifications. However, the paper should not be limited to comparing the results of experiments and calculations. It would be very meaningful to discuss the causes of the differences in the results of different samples, which is important for the formulation and design of materials. Furthermore, some information in the Results and Discussion section is a bit unclear. All the minor and major weaknesses are discussed in more detail in the following paragraphs.

Point 1: The development of green composite materials is introduced. Compared with traditional chemical composites, what are the advantages of fiber reinforced composites and biological composites? Especially in engineering applications? It would be better to provide typical application scenarios of green composite materials.

Point 2: On line 116, what is the difference between an open section and a closed section? And what are their different uses?

Point 3: The authors introduced the research status in detail, including various theoretical methods, which is valuable. It is suggested to list a separate paragraph in the introduction section to summarize the advantages and disadvantages of each method and the scope of application.

Point 4: Please explain the meaning of PLA where it first appears in the text.

Point 5: On line 216, I suggest that the subgraphs of Figure 3 be numbered with a, b, and c, unified as other figures.

Point 6: On line 220. the unit should be “g/m2”.

Point 7: On line 268, if no clamping device is used to limit the rotation of the sample, is torque be generated due to its own rotation? Does it affect the experiment? Experimental verification is not necessary, if possible, I hope the author can make appropriate analysis and explanation.

Point 8: And, on figure 9, lateral bulking appears of C180. Is it caused by rotation?

Point 9: On page 13, line 31, The scattering of the experiment is mentioned several times in the paper. It is suggested that the author introduce more factors affecting the scattering in the introduction.

Point 10: On page 13, the line number of the document starts from 1 again. Please correct it.

Point 11: On page 15, line 61, “…FEM model were modified to clamped-clamped…” This statement is strange because there is no clamping device used in the experimental setup, Line 268.

Point 12: On page15, line 76, “…with a slenderness ratio between 48 and 52, are out of the 76 limits of the modified Ludwick law method…”. The authors constructed three samples, the slenderness ratio of C60 is around 375, and C90 is around 170, C180 is around 50. Although the results in this paper show that the modified Ludwick law method is not suitable for C180, no more precise critical conditions have been found. For example, I mean if possible, the authors can set C120, C150 and other intermediate conditions, unlike the existing three samples, where the slenderness ratio gap is too large.

Reviewer 2 Report

The manuscript Prediction of critical buckling load on open cross-section columns of flax/PLA green composites is discusing about buckling load by diferent coomposites. It is overall written very well. Therefore, I recommend its acceptance after minor revisions.

There are plenty of paragraphs on the each page. At least 7-8 setences could be merges as a paragraph rather than each sentence.

More litrature could be added with recent references. 

Comparitive study should be made as a case one table could be added.

Reviewer 3 Report

Even the article is organized well and has some novelty,

1. It contains repeated section (2.2 theoretical approach which had been discussed before in recent work by authors 

Jiao-Wang, L., Larriba, C., & Santiuste, C. (2023). On the experimental validation of Ludwick law to predict critical buckling load of nonlinear elastic columns. Composite Structures303, 116237.

2. About 30% self citation is noticed. (around 6-7 articles from 22 references) 

Therefore, I suggest to modify all of these point and resubmit the article to the journal. 

Round 2

Reviewer 1 Report

Accept in present form

Reviewer 3 Report

The paper can be accepted in the current form.